# Defining the *Rhizobium leguminosarum* Species Complex

**DOI:** 10.3390/genes12010111

**Published:** 2021-01-18

**Authors:** J. Peter W. Young, Sara Moeskjær, Alexey Afonin, Praveen Rahi, Marta Maluk, Euan K. James, Maria Izabel A. Cavassim, M. Harun-or Rashid, Aregu Amsalu Aserse, Benjamin J. Perry, En Tao Wang, Encarna Velázquez, Evgeny E. Andronov, Anastasia Tampakaki, José David Flores Félix, Raúl Rivas González, Sameh H. Youseif, Marc Lepetit, Stéphane Boivin, Beatriz Jorrin, Gregory J. Kenicer, Álvaro Peix, Michael F. Hynes, Martha Helena Ramírez-Bahena, Arvind Gulati, Chang-Fu Tian

**Affiliations:** 1Department of Biology, University of York, York YO10 5DD, UK; 2Department of Molecular Biology and Genetics, Aarhus University, 8000 Aarhus, Denmark; sm.moeskjaer@gmail.com; 3Laboratory for Genetics of Plant-Microbe Interactions, ARRIAM, Pushkin, 196608 Saint-Petersburg, Russia; AAfonin@ARRIAM.ru; 4National Centre for Microbial Resource, National Centre for Cell Science, Pune 411007, India; praveen@nccs.res.in; 5Ecological Sciences, The James Hutton Institute, Invergowrie, Dundee DD2 5DA, UK; Marta.Maluk@hutton.ac.uk (M.M.); Euan.James@hutton.ac.uk (E.K.J.); 6Department of Ecology and Evolutionary Biology, University of California, Los Angeles, CA 90095, USA; izabelcavassim@gmail.com; 7Biotechnology Division, Bangladesh Institute of Nuclear Agriculture (BINA), Mymensingh 2202, Bangladesh; mhrashid08@gmail.com; 8Ecosystems and Environment Research Programme, Faculty of Biological and Environmental Sciences, University of Helsinki, FI-00014 Helsinki, Finland; aregu.aserse@helsinki.fi; 9Department of Microbiology and Immunology, University of Otago, Dunedin 9016, New Zealand; benjamin.perry@postgrad.otago.ac.nz; 10Departamento de Microbiología, Escuela Nacional de Ciencias Biológicas, Instituto Politécnico Nacional, Ciudad De México 11340, Mexico; entaowang@yahoo.com.mx; 11Departamento de Microbiología y Genética, Universidad de Salamanca, Instituto Hispanoluso de Investigaciones Agrarias (CIALE), Unidad Asociada Grupo de Interacción planta-microorganismo (Universidad de Salamanca-IRNASA-CSIC), 37007 Salamanca, Spain; evp@usal.es (E.V.); raulrg@usal.es (R.R.G.); 12Department of Microbial Monitoring, ARRIAM, Pushkin, 196608 Saint-Petersburg, Russia; eeandr@gmail.com; 13Department of Crop Science, Agricultural University of Athens, Iera Odos 75, Votanikos, 11855 Athens, Greece; tampakaki@aua.gr; 14CICS-UBI—Health Sciences Research Centre, University of Beira Interior, 6201-506 Covilhã, Portugal; jdflores@usal.es; 15Department of Microbial Genetic Resources, National Gene Bank (NGB), Agricultural Research Center (ARC), Giza 12619, Egypt; samehheikal@hotmail.com; 16Institut Sophia Agrobiotech, UMR INRAE 1355, Université Côte d’Azur, CNRS, 06903 Sophia Antipolis, France; marc.lepetit@inrae.fr; 17Laboratoire des Symbioses Tropicales et Méditerranéennes, UMR INRAE-IRD-CIRAD-UM2-SupAgro, Campus International de Baillarguet, TA-A82/J, CEDEX 05, 34398 Montpellier, France; stephane.boivin@cirad.fr; 18Department of Plant Sciences, University of Oxford, Oxford OX1 3RB, UK; beatriz.jorrin@plants.ox.ac.uk; 19Royal Botanic Garden Edinburgh, 20A Inverleith Row, Edinburgh EH3 5LR, UK; gkenicer@rbge.org.uk; 20Instituto de Recursos Naturales y Agrobiología de Salamanca (IRNASA-CSIC), Unidad Asociada Grupo de Interacción Planta-Microorganismo (Universidad de Salamanca-IRNASA-CSIC), 37008 Salamanca, Spain; alvaro.peix@csic.es; 21Department of Biological Sciences, University of Calgary, 2500 University Drive NW, Calgary, AB T2N 1N4, Canada; hynes@ucalgary.ca; 22Departamento de Didáctica de las Matemáticas y de las Ciencias Experimentales. Universidad de Salamanca, 37008 Salamanca, Spain; mh.ramirezb@usal.es; 23Microbial Prospection, CSIR-Institute of Himalayan Bioresource Technology, Palampur (H.P.) 176 061, India; gal_arvind@yahoo.co.in; 24State Key Laboratory of Agrobiotechnology, Rhizobium Research Center, and College of Biological Sciences, China Agricultural University, Beijing 100193, China; cftian@cau.edu.cn

**Keywords:** *Rhizobium*, species complex, bacterial taxonomy, core genes, housekeeping genes, average nucleotide identity, speciation, genospecies

## Abstract

Bacteria currently included in *Rhizobium leguminosarum* are too diverse to be considered a single species, so we can refer to this as a species complex (the Rlc). We have found 429 publicly available genome sequences that fall within the Rlc and these show that the Rlc is a distinct entity, well separated from other species in the genus. Its sister taxon is *R. anhuiense*. We constructed a phylogeny based on concatenated sequences of 120 universal (core) genes, and calculated pairwise average nucleotide identity (ANI) between all genomes. From these analyses, we concluded that the Rlc includes 18 distinct genospecies, plus 7 unique strains that are not placed in these genospecies. Each genospecies is separated by a distinct gap in ANI values, usually at approximately 96% ANI, implying that it is a ‘natural’ unit. Five of the genospecies include the type strains of named species: *R. laguerreae, R. sophorae, R. ruizarguesonis*, *“R. indicum”* and *R. leguminosarum* itself. The 16S ribosomal RNA sequence is remarkably diverse within the Rlc, but does not distinguish the genospecies. Partial sequences of housekeeping genes, which have frequently been used to characterize isolate collections, can mostly be assigned unambiguously to a genospecies, but alleles within a genospecies do not always form a clade, so single genes are not a reliable guide to the true phylogeny of the strains. We conclude that access to a large number of genome sequences is a powerful tool for characterizing the diversity of bacteria, and that taxonomic conclusions should be based on all available genome sequences, not just those of type strains.

## 1. Introduction

The increasing availability of genome-scale DNA sequencing is transforming the practice of bacterial taxonomy. Until recently, bacterial species have been described using an eclectic mixture of phenotypic characteristics plus a limited amount of DNA-based information—an approach called ‘polyphasic taxonomy’ [1,2]. Over time, the DNA component has come to be considered the most critical information, with the sequence of small subunit ribosomal RNA (16S rRNA) providing reliable placement down to the genus level, and DNA–DNA hybridization (DDH) used to distinguish species [3,4,5]. Most journals that publish new species descriptions now require a genome sequence for the proposed type strain, and there are initiatives to provide genome sequences for species described in the past [6,7,8]. These developments recognize the power of genome information [9,10,11,12,13,14], although genome sequences are not a formal requirement of the code for bacterial nomenclature [15]. So far, many authors have simply used the genome sequence to provide average nucleotide identity (ANI) values that are a more convenient and accurate substitute for the outdated DNA–DNA hybridization (DDH) laboratory technique [16,17], and perhaps to extract the sequence of 16S rRNA and a few housekeeping genes that would otherwise have required separate amplification and sequencing. These are used for comparison to related species, but usually only to the type strains, i.e., a single strain representing each named species. Genome sequences are capable of providing much more information though, particularly if they are available for multiple strains of a species and not just for the type strain. They allow core genes, which are present in all strains and experience relatively limited lateral gene transfer, to be distinguished from accessory genes, which are more sporadic in distribution, often transferred, and responsible for many of the phenotypic differences among strains and species [18]. Multiple core genes can be used to construct very robust phylogenies because discrepancies affecting individual genes are averaged out [19,20], while the distribution of accessory genes may also help to distinguish species [21].

In 1987, a committee for bacterial systematics declared that “the complete deoxyribonucleic acid (DNA) sequence would be the reference standard to determine phylogeny and that phylogeny should determine taxonomy” [4]. It would be another eight years before the first bacterial genome was sequenced [22], so the committee could only recommend DDH as the best available substitute at the time, and defined a genospecies as including those strains with approximately 70% or greater DDH relatedness and less than 5 °C melting temperature difference. It is essentially in this sense that we use the term ‘genospecies’, although of course we can now use complete DNA sequences to assess relatedness much more accurately and in multiple ways. In fact, the term ‘genospecies’ had already been defined for bacteria by Ravin in 1963: “When their respective genotypes permit inter-bacterial genetic transfer and recombination, we may say that they belong to the same genospecies” [23]. This is closer to the widely used biological species concept as defined by Mayr [24], and suggests that recombination could provide genetic cohesion within a species. It is known that sequence divergence can prevent homologous recombination, so diverged sequences will continue to diverge further, and this could create genetic barriers between species [25,26]. While natural selection and ecology are undoubtedly also important in bacterial speciation, it is indeed observed that, in many bacterial groups, there are ‘gaps’ in genome-based distance measures, approximately at 70% DDH or 95–96% ANI, that allow the definition of species without too many ambiguous intermediates [27,28]. For our purposes here, we can say that a genospecies is a discrete cluster in the sequence space of core genes [29].

There have been some broad-scale initiatives to establish taxonomic databases for all bacteria and archaea based on genome sequences. The Type Strain Genome Server (TYGS) includes only type strains, but provides tools to compare any query genome with the most relevant type strains using a whole-genome similarity metric called dDDH, which is designed to correlate well with the laboratory measurements of DDH that were used in the past [30]. On the other hand, the Genome Taxonomy Database (GTDB) uses a 95% ANI threshold to define species and aims to include and classify all available genomes [20], including many that do not fit into species that have been formally described. By necessity, the project generates temporary names to accommodate these genomes. These initiatives are very welcome and potentially provide a universal framework, but the proof of their utility will come from thorough investigations at a much finer-grained level.

In the present study, we explore the potential of genome sequence data to illuminate the diversity of an important and well-studied bacterial group, *Rhizobium leguminosarum* in the broad sense, the archetype of rhizobia. Rhizobia are bacteria that induce the formation of nodules on the roots of legume plants (Fabaceae). The rhizobia colonize cells within the nodule and use energy supplied by the plant to fix nitrogen (i.e., reduce atmospheric N_2_ to ammonia, NH_3_) and make organic nitrogen compounds available to the plant. This is a mutually beneficial symbiosis, and its importance for agriculture has stimulated research for more than a century [31,32].

The name *Rhizobium leguminosarum* was proposed by Frank in 1889 for these root nodule bacteria [33]. In those early days, there was considerable confusion about their affinities, or even whether they were bacteria at all, but by 1926 the taxonomy had settled down and a number of species were recognized on the basis of host specificity and cultural properties: *R. leguminosarum* and *R. japonicum* [34], and *R. meliloti*, *R. trifolii* and *R. phaseoli* [35]. The taxonomy remained stable until 1982, when a new species, *R. loti*, was described [36] and, in the same year, *R. japonicum* was considered sufficiently distinct to merit a new genus, *Bradyrhizobium* [37]. Eventually, *R. meliloti* and *R. loti* were also transferred to new genera, *Sinorhizobium* and *Mesorhizobium* [38,39]. On the other hand, in the 1984 Bergey’s Manual, Jordan amalgamated *R. trifolii* and *R. phaseoli* into *R. leguminosarum* because the only consistent difference among the three was their host specificity [40], which was plasmid-encoded and could be transferred among strains [41,42,43]. Jordan proposed three biovars, *viciae*, *trifolii* and *phaseoli*, to accommodate the three distinct host-specificity types within *R. leguminosarum* [40]. Since then, there has been a proliferation of new species described within the genus *Rhizobium*, as well as new genera for more distant species of root-nodule bacteria [44]. Many strains originally described as *R. leguminosarum* have been moved to new species, and it is clear that the three biovars, now called symbiovars [45], are not confined to *R. leguminosarum*, because almost identical nodulation genes (host-specificity determinants) can be found in other *Rhizobium* species [46]. Ramírez-Bahena et al. [47] reconsidered the taxonomic status of *R. leguminosarum* with the benefit of gene sequence data and concluded that, while *R. trifolii* was indeed a synonym of *R. leguminosarum*, the type strain of *R. phaseoli* defined a separate species (although some other strains of symbiovar *phaseoli* do belong to *R. leguminosarum*). They also discovered that two different bacteria were being distributed as the type strain of *R. leguminosarum*, decided which of these was the true USDA 2370^T^, and defined the other (DSM 30132^T^) as the type of a new species, *R. pisi*.

Despite the description of many new species, the genetic diversity that remains within *Rhizobium leguminosarum*, as currently defined, is still greater than is considered typical for a single bacterial species. Kumar et al. [29] noted that the genome-based diversity of *R. leguminosarum* isolates from legume nodules at a single site was ten times higher than that of *Sinorhizobium medicae* from the same site [48]. They divided the isolates into five genospecies (A to E) that were separated by ANI values below 95%, a level that is widely considered to be evidence that they represent separate species [49]. They also considered the small number of other published *R. leguminosarum* genomes available at that time, and found that some could be accommodated within the five genospecies, while others potentially represented additional genospecies. A larger survey of almost 200 genomes from three north European countries found only the same five genospecies [21], but another study with genomes from a wider European geographic range included some that fell into two novel genospecies that were closely related to each other [50]. The authors called these gsF-1 and gsF-2, and the second of these included strain FB206T, the type strain of *R. laguerreae*, a species that was described in 2014 [51]. This description was based on six isolates that were distinct from USDA 2370^T^, the type strain of *R. leguminosarum*, in housekeeping gene sequences and by DNA–DNA hybridization, even though their 16S rRNA sequences were all identical to that of USDA 2370^T^. As measured by ANI, strains of gsB and gsC are just as divergent from USDA 2370^T^ as are those of *R. laguerreae* [50], so it seems likely that these genospecies could also be described formally as distinct species using conventional taxonomic criteria. The type strain USDA 2370^T^ is in gsE, so this genospecies remains *R. leguminosarum* in the narrow sense.

Recently, several published [21,50,52] and unpublished [53] studies have augmented the public databases with a large number of new genome sequences assigned to, or related to, *R. leguminosarum*. In addition, there have been numerous contributions of individual genomes [54,55,56,57,58,59,60,61,62]. Here, we examine this wealth of genomic information in order to establish whether *R. leguminosarum*, in the broad sense, is a distinct entity, and whether it can be subdivided into clearly defined groups that might, in future, be named as separate species within the overall *R. leguminosarum* species complex (Rlc).

## 2. Materials and Methods

### 2.1. The Set of Genome Sequences

All genome sequences for the genus *Rhizobium* (NCBI:txid379) were downloaded in FASTA format from the NCBI database (www.ncbi.nlm.nih.gov) using the script genbank_get_genomes_by_taxon.py that is distributed as part of the pyani package (http://widdowquinn.github.io/pyani/). Initial analysis was based on the 834 genomes available on 25 July 2020 (Appendix A). All later analyses were confined to the genomes that belonged to the Rlc or to its sister taxon *R. anhuiense*, using the genomes that were available on 28 August 2020. After eliminating duplicate sequences of the same strain, there were 429 genomes in the Rlc and 11 in *R. anhuiense* (440 genomes altogether). These are listed in Appendix A, together with additional relevant information. Files were renamed to include the strain name as well as the accession name, and also the known genospecies in the case of the strains previously described by Cavassim et al. [21].

### 2.2. The Core Gene Phylogeny

A core gene phylogeny was constructed from the bac120 set of 120 universal bacterial core genes used by Parks et al. [63]. Whereas Parks et al. used protein sequences for a phylogeny encompassing all bacteria, we used the DNA sequences to gain maximum resolution of closely related strains. The list of protein IDs in Appendix A of Parks et al. [63] was used to identify these proteins in the annotated complete genome sequence of WSM1325 (GCF_000023185.1; [64]), or another Rlc strain if not annotated in this genome (Appendix A), and the set of protein sequences from this strain was used to find the corresponding genes in each genome using TBLASTN [65] with an E-value cutoff of 1E-10. Hits were extended at each end by the expected number of nucleotides to obtain the full gene sequence, or to the end of the contig, if nearer. All 120 genes were located in the chromosome in the fully assembled genome of strain 3841 [66]. Sequences for each gene were aligned using clustalo 1.2.3 [67], and then all 120 genes were concatenated before constructing an approximately maximum likelihood phylogeny using FastTree 2.1.10 [68], with local bootstrap branch support values. Of the initial 834 genomes (Appendix A), 37 lacked some or all of the core genes (Appendix A; presumably these assemblies were incomplete), while 797 had all 120 genes and were included in Figure 1. A new alignment was constructed for the 440 focal genomes (Rlc plus *R. anhuiense*) and an initial phylogeny constructed using FastTree (Appendix A). Although FastTree is very fast, other algorithms have been shown to find trees of higher likelihood [69], so the phylogeny was subsequently refined using RAxML-NG 1.0.1 [70] on the CIPRES Science Gateway at https://www.phylo.org [71], after selecting the optimal model (GTR + FC + I + G4 m) using ModelTest-NG 0.1.3 [72]. Nonparametric bootstrapping (allowing up to 1000 replicates) converged after 300 replicates. The trees were displayed using Dendroscope 3.7.2 [73] for initial exploration, FigTree 1.4.3 (https://github.com/rambaut/figtree/releases) for Figure 1, and iTOL [74] for other figures. Python scripts are available at https://github.com/jpwyoung/Rlc and the iTOL trees at https://itol.embl.de/shared/rhizobium.

### 2.3. Average Nucleotide Identity (ANI)

Pairwise ANI was calculated using fastANI 1.31 [75] with the default settings (kmer = 16, fragment length = 3000, minimum shared fraction = 0.2), and displayed using the Seaborn library with custom Python scripts available at https://github.com/jpwyoung/Rlc. Within the Rlc plus *R. anhuiense*, the number of sequence fragments per genome ranged from 2189 to 3829, and the number matched in pairwise comparisons ranged from 1640 to 3825.

### 2.4. Analysis of 16S and Nodulation Genes

The 16S rRNA sequence of USDA 2370^T^ was used as the query to find the most similar sequence in each genome using BLASTN (default parameters). To assign the symbiovar, NodD, NodA, NodB and NodC protein sequences of strains representing the three symbiovars (3841 for *viciae*, WSM1325 for *trifolii*, *R. etli* CFN42 for *phaseoli*) were used as queries to search each genome using TBLASTN (E-value 1E-5) to find the corresponding genes, and the symbiovar determined by the best match. Each of the four genes gave the same result, except that a few SEMIA strains had 100% match to the CFN42 NodA sequence, but no match to NodB, NodC or NodD, so were recorded as non-nodulating. The 16S and *nodC* sequences were extracted and aligned for phylogeny as described for core genes. Three assemblies had no 16S sequence (128C53, Ps8, Vh3), and a further three (FB403, SP4, FA23) were omitted because their 16S sequence was incomplete.

### 2.5. Housekeeping Genes for Genospecies Assignment

Three housekeeping genes that have been widely used to characterize isolates, *recA* [76], *atpD* [76] and *gyrB* [77,78,79], were located in each genome by TBLASTN (E-value 1E-10) using the corresponding protein sequences annotated in an arbitrary Rlc strain (SM130B, gsA). To simulate the procedure of a typical application, the gene sequences were trimmed to the expected informative part of the amplicon (excluding primers) and aligned separately using ClustalW [80] before concatenation in the order *atpD-gyrB-recA* and phylogeny reconstruction with RAxML-NG, as for core genes (above).

Sequences corresponding to the shorter amplicons used by Fields et al. [81] for high-throughput sequencing of parts of *recA* and *rpoB* from bulk nodule DNA were located by BLASTN (E-value 10) using sequences of strain SM3 (gsB) from [81] and aligned with clustalo for phylogeny with RAxML-NG.

### 2.6. ANI of Chromosomal and Plasmid Compartments

The scaffolds of each genome assembly were separated into chromosomal and nonchromosomal (plasmid) compartments. Scaffolds were classified as chromosomal if they carried one or more of the 3215 genes identified by Cavassim et al. [21] as normally chromosomal in Rlc genomes, as assessed by BLASTN (E-value 1E-10, only the best hit in each genome). The average nonchromosomal compartment was 2.496 Mb (range 0.957–4.436 Mb), making up 32.8% (12.6–47.0%) of the total genome. The average chromosomal compartment was 5.082 Mb (range 4.591–6.923 Mb). We tested a more stringent definition of chromosomal scaffolds, requiring at least five of the genes, which resulted in an average chromosomal compartment of 5.033 Mb. As this is only 1% lower, we concluded that our original classification was sufficiently robust. ANI was calculated separately for the two compartments using fastANI 1.31 [70] and displayed using the Seaborn library with custom Python scripts available at https://github.com/jpwyoung/Rlc.

### 2.7. Identification and Analysis of Ortholog Sets

The aim here was to assess the degree to which accessory genes were shared between strains of the same or different genospecies. Putative genes were identified in each genome assembly using Prodigal 2.6.3 [82]. The predicted proteins were sorted into groups of orthologs using Orthofinder 2.4.0 [83]. The number of orthogroups shared between each pair of genomes, and the total number of orthogroups in each genome, were extracted from the output file Orthogroups.GeneCount.tsv. Genes in single copy in a single genome were not included. Orthogroups with more than two copies in any one genome were excluded, as these often had ambiguous or obscure orthology. A normalized gene sharing index between two strains was calculated as the average fraction of the orthogroups present in a strain that were shared with the other strain. The computing for this part of the project was performed on the Aarhus University GenomeDK cluster. It was not computationally feasible to run Orthofinder on the full set of 440 genomes, so the Rlc was split into four major clades (gsL + M + C, D + E, H + A, the rest) and each was run separately. In addition, a set of 100 genomes that included examples from each genospecies, and nine additional smaller sets, were used to sample values for more distantly related strain pairs. Across all runs, the average number of orthogroups was 12,829 (range 10,314–16,965) before filtering on copy number, and 12,198 (range 10,019–15,811) after filtering.

## 3. Results and Discussion

### 3.1. The Genus Rhizobium and Related Genera

All genomes identified as the genus *Rhizobium* and available at NCBI were downloaded on 25 July 2020 (834 genomes, Appendix A). The sequences of 120 core genes were extracted from each genome assembly. The full set of core genes was found in most assemblies, but 37 were incomplete (Appendix A). A phylogeny (not shown) based on all assemblies that were complete enough to have at least 100 of the genes showed that GCF_003001755.1, annotated as *R. tropici* NFR14, was an outlier. Analysis of this assembly by the TYGS server (https://tygs.dsmz.de/) indicated that it actually belonged to the genus *Bradyrhizobium*, so this was used as an outgroup to root the tree, together with *R.* sp. FKL33, a distant member of the *Rhizobiaceae* according to the Genome Taxonomy Database (GTDB, https://gtdb.ecogenomic.org). The tree for genomes that had a complete set of 120 core genes is shown in Figure 1.

**Figure 1 genes-12-00111-f001:**
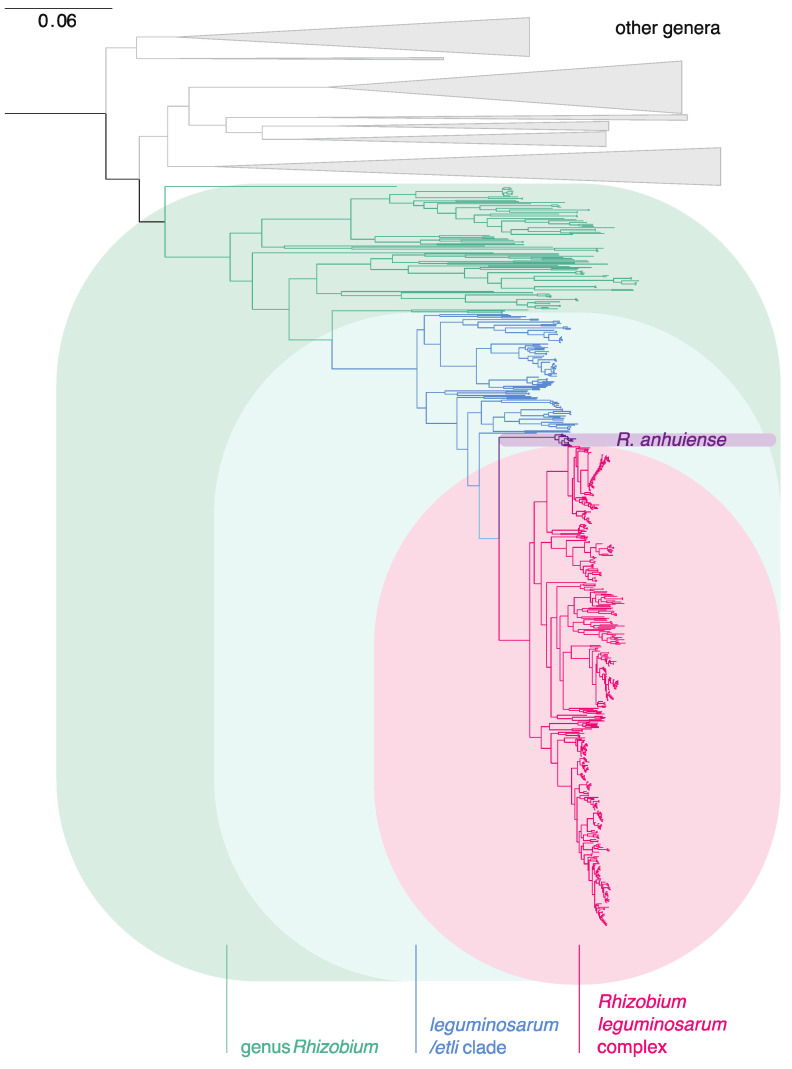
Phylogeny, based on 120 core genes, of genomes assigned to the genus *Rhizobium* by NCBI (NCBI:txid379). Only the 797 genomes that had all 120 genes were included. Genomes that are not currently included in the genus *Rhizobium* are indicated in the basal clades, collapsed by genus: *Pararhizobium, Mycoplana, Agrobacterium,* GTDB g__Rhizobium_A [20], *Pseudorhizobium, Neorhizobium,* and *Allorhizobium* (top to bottom). The tree was rooted using strain FKL33 (not shown).

Many strains in the basal clades of this phylogeny have featured in previous studies that classified them in genera related to, but distinct from, *Rhizobium*, and this allowed us to identify the main clades as *Allorhizobium [84,85]*, *Neorhizobium [85,86]*, *Pseudorhizobium [87,88]*, *Agrobacterium [89]*, *Mycoplana [90]*, *Pararhizobium [85]*, and an unnamed genus-level clade identified in GTDB as g__Rhizobium_A, which includes the type strains of *Rhizobium rhizoryzae* and *Allorhizobium pseudoryzae*. It should be emphasized that the only strains of these genera that are shown in Figure 1 are those that are misclassified as *Rhizobium* in the NCBI database. The database also includes a much larger number of related strains that are correctly classified. The systematic use of ANI can be expected to improve the NCBI classification over time [91]. In Appendix A, we list the genera to which these strains appear to belong. Clearly, there is scope to use genome sequences to improve our understanding of all these genera, but that is outside our present focus.

The genus *Rhizobium*, as currently understood, is separated by a long branch from other genera. Its sister clade is a single strain, “*R. album*” NS-104^T^ [92], that has no sequenced relatives and is not clearly affiliated with any known genus. Although described as a species within *Rhizobium*, its distance from all other species in the genus (Figure 1) suggests that this needs reconsideration. The genomes that are in *Rhizobium* but not in the Rlc are listed in Appendix A. Within *Rhizobium*, a very long branch distinguishes a clade that includes *R. leguminosarum*, *R. etli*, and a number of more recently described species, including *R. laguerreae, sophorae, indicum, ruizarguesonis, anhuiense, ecuadorense, acidisoli, chutanense, hidalgonense, vallis, pisi, fabae, bangladeshense, sophoriradicis, phaseoli, esperanzae, aethiopicum, aegyptiacum*, *binae* and *lentis*.

### 3.2. The Rlc has a Clear Boundary

The group that we identify as the Rlc forms a distinct subclade within the *leguminosarum-etli* clade, and its sister taxon is *R. anhuiense* (Figure 1). The clear boundary between the Rlc and other *Rhizobium* species was further supported by an analysis of genome-wide ANI, using the type strain of *R. leguminosarum* (USDA 2370^T^) as the reference (Figure 2). All genomes within the Rlc have ANI values of 92.15% or higher with respect to USDA 2370^T^. Values for *R. anhuiense* are tightly grouped in the range 91.33–91.55%, while the rest of the *leguminosarum-etli* clade starts at 90.18% and ranges down to 86.68%. After that, there is a large drop in ANI to *R. alamii* (83.16%), the sister taxon of the *leguminosarum-etli* clade, reflecting a long branch in the tree. There are no further breaks in ANI, even at the boundary of the genus *Rhizobium*. Indeed, there is some overlap in ANI values between comparisons within *Rhizobium* and those with related genera. This seems a little surprising, considering that the phylogeny shows a long, well supported branch at the base of the genus, but ANI (and especially FastANI) is not very sensitive at values below 80% [75]. Overall, there is very good agreement between the boundaries seen in the phylogeny and those detected by ANI, and there is no ambiguity about which genomes belong within the Rlc and which are outside it. However, since strains within the Rlc can have ANI values as low as 92.15% with the type strain of *R. leguminosarum*, it is evident that the diversity within the Rlc is greater than can be encompassed within a single bacterial species (for which a threshold is usually considered to be approximately 95–96%). An important question is whether the structure of the Rlc is amorphous or can be broken down into a set of clearly defined species-level units. That is the main goal of our study.

There were genomes of 429 strains within the Rlc available from NCBI by 28 August 2020, and these genomes, together with the 11 genomes of *R. anhuiense* as an outgroup, were used for all subsequent analyses. They are listed in Appendix A, while Appendix A lists a few additional Rlc genomes that were not used because they were duplicates.

### 3.3. Genospecies Can Be Defined within the Rlc

A core gene phylogeny of the Rlc and its sister clade *R. anhuiense* is shown in Figure 3. The Rlc is highly diverse, with many well-supported clades at various depths. To help us to decide which clades were sufficiently distinct to be considered genospecies, we also considered ANI values. In Figure 4a, which shows ANI values for all pairwise comparisons among the 440 strains, the five genospecies (A–E) defined previously [29] are clearly visible as red squares (ANI > 96%). There are many strains that fall outside these five genospecies, and most of them are also in clusters that are potential new genospecies (identified by the letters G–S) because within-cluster ANI values are above 96%.

Published studies of ANI usually suggest a threshold for species designation of 95–96% [16,17,27]. To see the effect of these thresholds, the ANI data of Figure 4a are plotted again in Figure 4b with cutoffs at 95% and 96%. For most clades, there is no ambiguity if a 96% threshold is used. The exceptions are the two strains in genospecies J (gsJ), which have ANI values above 96% with some, but not all, strains of gsB, and the clade that includes gsN, O, P, Q and R, which are all closely related. Collectively, we call this the F-clade, since two potential new genospecies in this group were previously called F-1 and F-2 [50]. Our new analysis suggests that there are several genospecies in this group so, to avoid confusion, we have not used F as a genospecies name. Lowering the threshold to 95% (grey areas in Figure 4b) would lead to increased ambiguity for some of the genospecies.

Altogether, a 96% ANI threshold allows us to define 18 potential genospecies within the Rlc that each have at least two genome sequences, plus 7 single strains that have no close relatives. If we want to define genospecies that reflect real biological units, we have to recognize that a single arbitrary threshold for ANI may not be applicable. We need to assess each potential group for coherence and distinctness. For each of our proposed genospecies, we selected a representative strain, using the type strain where taxonomic species have already been defined, or a representative strain that has been well studied, has a good-quality genome, and is reasonably central within the genospecies. Using each of these 18 strains in turn as the reference, ANI values for all 440 strains are plotted in the panels of Figure 5. These plots provide justification for each of the genospecies we are proposing. In each plot, there are gaps in the ANI values, including a gap at approximately 96% ANI that we take to be the boundary of the genospecies. This gap is sometimes very large, e.g., more than 2 percentage points for gsE (between 95.00% and 97.40%), but small for some species in the F-clade, e.g., the gap for gsO is between 95.96% and 96.30%.

Genospecies vary in their compactness: all gsB strains have ANI values above 98% (not only with the reference strain, but also in all pairwise combinations), whereas values for gsA and gsC extend closer to 96%. We consider that the compactness of gsB justifies keeping gsJ as a separate species, even though it currently has just two strains and their ANI to some gsB strains is slightly above 96%, because it is likely that all the gsB strains share a large set of characteristics that may not be present in gsJ.

The F-clade (gsN, O, P, Q, R) is a part of the Rlc that seems to have a less clearly defined structure than the rest. Most, though not all, of the pairwise ANI values within the F-clade are above 95% (Figure 4b). At a 96% threshold, the five proposed genospecies are generally well defined. However, there are two strains, SPF2A11 and HP3, that have ANI values above 96% to all members of gsQ, but also to most members of gsR (visible as black bars in Figure 4b). We have assigned them to gsQ on the basis of the phylogeny (Figure 3). *R. laguerreae sensu stricto* (gsR) is very compact, with all ANI values above 98%, so we have not included the related strain CCBAU10279 within it, even though its ANI to the type strain is 96.54%.

In the core gene phylogeny (Figure 3), every genospecies is a clade with 100% local bootstrap support for its subtending branch, except gsP, which is not monophyletic in the RAxML-NG tree (Figure 3) and has a support of just 30.3% in the FastTree tree (Appendix A). There are only two genomes in gsP, Vaf10 and Vaf108, and it is notable that they have long individual branches in the phylogeny, although their pairwise ANI is 96.25%, which is consistent with including them in the same species and higher than their ANI with any other genome.

The core gene phylogeny is based on a set of 120 genes. To test its robustness, we split the genes arbitrarily (by TIGR number, Appendix A) into two independent sets of 60 genes and repeated the analysis. The resulting phylogenies (Appendix A) are highly congruent with the original phylogeny. All 18 genospecies are supported by 100% bootstraps, except that gsP has just 53% support in the set B tree and the two strains of gsP in are not grouped together in set A, with Vaf108 becoming internal to gsO. This is further evidence that the putative gsP (Vaf10 and Vaf108) is not a robust grouping. The relationships among the other genospecies are conserved, apart from a minor difference in the placement of gsK, which has poor bootstrap support in all three trees. We can conclude that the set of 120 core genes is a sufficiently large sample to provide a reliable phylogeny for defining the genospecies and their relationships.

Out of the 429 Rlc genomes currently available, just seven do not fit into any of the 18 genospecies that we have defined: CC278f and Norway that are related to gsD, the deeply branching WYCCWR10014, Tri-43 related to gsS and gsG, WSM1689 and CCBAU10279 related to gsR, and Vaf12 related to gsK. Apart from CCBAU10279 (discussed above), none of these have ANI above 96% with any other genome. It is possible that each of these is the first sequenced example of a novel genospecies, but there might also be technical reasons for their divergence from other genomes, so they are best left until additional examples are discovered.

Five of the genospecies include the type strains of species whose names have been effectively published. Genospecies E includes *R. leguminosarum* USDA 2370^T^, so this genospecies is the species *R. leguminosarum* in the narrow sense (*sensu stricto*). Similarly, gsR is *R. laguerreae*, gsG is *R. sophorae*, gsC is *R. ruizarguesonis*, and gsI is *“R. indicum”*. In each case, we have used the published type strain as the representative of the genospecies in our analyses.

### 3.4. The Genospecies Are Consistent with Genomic Taxonomy Databases

When the genomes of the 18 strains representing the proposed genospecies were submitted to TYGS [30], those representing gsC, gsE, gsG, gsI and gsR were correctly identified as *R. ruizarguesonis, R. leguminosarum, R. sophorae, “R. indicum”* and *R. laguerreae*, respectively, while the remaining 13 genomes were all described as ‘potential new species’ (Appendix A). Furthermore, the similarity score dDDH(d_4_), which should be above 70% for conspecific strains, was no greater than 66.1% for any of the pairwise comparisons among the 18 representative strains, supporting our proposition that they represent 18 distinct genospecies. All of the seven unique strains that did not fall within these genospecies had dDDN(d_4_) values below 70% with the representative strains of their closest genospecies (Appendix A). Most were significantly below (65.8% or less), but CCBAU10279 scored 69.8% (C.I. 66.8–72.6%) against FB206, the type strain of *R. laguerreae* (gsR). This borderline value is consistent with the ANI value of 96.54 and our decision to leave this strain outside gsR despite this. Overall, then, we can conclude that our proposal to split the Rlc into 18 genospecies and 7 unique strains is completely supported by the dDDH-based species definition used by TYGS.

The Genome Taxonomy Database (GTDB) is another resource for genome-based taxonomy that provides a comprehensive genome-based taxonomy of all prokaryotes [20]. The database includes 287 of the 329 Rlc strains that we have studied, and they are assigned to ten species plus two single isolates. These species designations are consistent with the genospecies we have defined, but with some ‘lumping’ (Table 1), as can be expected because GTDB uses a fixed 95% ANI threshold. All strains in the F-clade (gsN, gsO, gsP, gsQ, gsR and two unique strains) are assigned to a single species, s__Rhizobium laguerreae. The isolates CC278f and Norway are included within gsD as s__Rhizobium leguminosarum_K. The related genospecies gsB and gsJ are combined in s__Rhizobium leguminosarum_L, together with FA23, the only representative of gsK in GTDB. The position of gsK is uncertain, with poor bootstrap support in both Figure 3 and Appendix A. The three small genospecies gsG, gsS and gsI do not yet have any representatives in GTDB. In other respects, the GTDB definitions of species within the Rlc are the same as ours.

### 3.5. The Genospecies Could Be the Basis for New Formal Taxonomic Names

Several valid species names already exist within the Rlc. The type strain of *R. leguminosarum*, USDA 2370^T^, is centrally placed within gsE (Figure 3), and this genospecies has a very clear boundary in the region of 96% ANI (Figure 5), so there is no doubt that gsE is synonymous with *R. leguminosarum sensu stricto*, providing a new, narrower definition of the species. Two much smaller clades are also readily equated with named species: gsG is *R. sophorae* [93] and gsI is “*R. indicum*” [59] (not yet a validated name). Another recent name is *R. ruizarguesonis* [58], described on the basis of four closely related strains, but clearly embedded within the large and diverse gsC, which we have concluded is best considered a single genospecies. Hence, gsC is *R. ruizarguesonis*, although the species description may need to be revisited to encompass this much greater diversity.

Finally, there is *R. laguerreae*, the first of these ‘new’ species to be named [51]. The description included six strains in this species, but no genome sequences were provided. Two strains have subsequently been sequenced, the type strain FB206^T^ and FB403; both are in gsR. The species description provides a phylogeny based on three housekeeping genes, *rpoB*, *recA* and *atpD*, which shows that three of the other strains are close enough that it is safe to conclude that they are also gsR. The sixth strain, CVIII4, was assigned to *R. laguerreae* because it had 82% DNA–DNA hybridization with the type strain. The evidence from housekeeping gene sequences is more ambiguous though. The *atpD* sequence places it closest to FB403 and HUTR05 in gsR, but the closest *rpoB* match is GLR2 (gsQ) followed by UPM1131 (gsO), while the *recA* sequence is highly divergent and closest to the three strains in gsL. This explains the diverged position of this strain in the concatenated phylogeny of the three genes, and illustrates once again the hazards of basing taxonomic assignments on single genes. In summary, it is not clear whether all the strains included in the description of *R. laguerreae* are in gsR. We have already commented that deciding on appropriate genospecies boundaries in the F-clade (gsN-R) is not straightforward. If genospecies are defined narrowly, as we have done, some pairs of strains in different genospecies have high similarity (>96% ANI), whereas if the whole F-clade is treated as a single species, many pairs are more distant than is usual within a species (<95% ANI). In the GTDB classification, based on a 95% ANI threshold, the whole F-clade is included in the species *R. laguerreae* [20]. It is evident from the phylogeny (Figure 3) that branch lengths are long in the F-clade, indicating a relatively rapid rate of evolution. The treatment of this clade is perhaps the most debatable taxonomic issue within the Rlc.

Some of the other genospecies seem much clearer candidates for the assignment of new species names. Genospecies B is notably compact, despite including strains from different studies, host plants and geographic areas. A well-known member is 3841, the first Rlc strain to have a published genome sequence [66]. Genospecies A is another clade with a clear boundary, though all the genomes are symbiovar *trifolii* so far, and the majority from a single study [21]. Genospecies D has eight genomes that are all very similar, despite diverse origins (Denmark, France and Australia), and they are well separated from anything else. There are other genospecies that also look clear cut, but so far they have fewer genomes, so their full extent may not have been sampled.

### 3.6. Housekeeping Gene Amplicons Can Identify Isolates to Genospecies

Most of our current knowledge of rhizobial diversity comes from studies in which substantial numbers of strains were isolated from root nodules. A fairly standard approach to characterizing these strains is to amplify and sequence part of the 16S rRNA gene and of a few (typically three) housekeeping genes. As discussed below, 16S sequence is not able to distinguish the genospecies of the Rlc, so we sought to establish whether individual housekeeping genes might be more informative.

We extracted, from the genomes, the sequences that would be expected using published primers for three commonly used housekeeping genes: *atpD, gyrB* and *recA*. The informative parts of these amplicons are 534, 719 and 602 bp in length, respectively. The concatenated sequence of all three genes was sufficient to classify strains of all 18 genospecies (Appendix A). Most genospecies formed a single clade, but gsO and gsP strains were scattered across the phylogeny. No exact concatenated sequence was shared between genospecies though. For a large strain collection, it would be convenient to sequence just a single gene, so the individual phylogenies are shown in Figure 6. Any of the three genes, considered singly, would suffice to identify most of the isolates correctly, but a few sequences would be ambiguous. For example, three strains of gsH (WSM1325, WSM1328 and WSM409) and three of gsC (SM47, SM49 and SM60) share an identical *recA* allele, WSM1481 (gsJ) shares an *atpD* allele with gsQ, and the Vaf10 (gsP) *gyrB* allele is embedded within gsN. In each case, there are also some alleles in different genospecies that are so close that new isolates with related sequences could not be classified with certainty. Overall, though, nearly all isolates could be classified correctly using just one housekeeping gene; ambiguous cases could then be resolved using additional genes.

Two further points of interest are evident in these single-gene phylogenies (Figure 6). The first is that some genospecies are not monophyletic in the single-gene trees, but have two or more distantly related alleles. For example, gsC has several distinct clades of *recA* sequences, including the one that is shared with some gsH strains. The implication is that these genospecies have experienced introgression of new alleles from other genospecies. Occasionally, this was a recent event and the alleles have not diverged, as in the gsC-gsH *recA* case, but most of the introgression appears to be much more ancient because alleles have diversified separately within each genospecies. The second point, somewhat related, is that each gene has a unique phylogeny. Furthermore, even if all three gene sequences are concatenated, the resulting tree (Appendix A) is significantly different from the reference phylogeny based on 120 genes (Figure 3). While most (though not all) of the genospecies form single clades, the relationships among genospecies are sometimes radically different. For example, the related genospecies gsC, gsL and gsM are very distant in the three-gene tree, as are the close pair gsD and gsE. This demonstrates that using just three genes does not provide sufficient resolution for a secure phylogenetic placement of potentially novel genospecies. For example, Youseif et al. [94] characterized a number of isolates using amplified sequences of the 16S, *glnA*, *rpoB* and *pgi* genes. Some of the isolates appear to be in or around the F-clade, but their exact identity remains uncertain.

As the technology becomes more accessible, whole-genome sequencing will probably replace the sequencing of single genes, even for large samples of isolates [21,50]. Single genes will, however, still be important for high-throughput amplicon sequencing (HTAS), an approach that characterizes the diversity of a microbial community by the diversity of sequences amplified from community DNA. While 16S rRNA is often used for general bacterial communities, protein-coding genes are more informative when a narrow taxonomic range is of interest. Partial sequences of *recA* and *rpoB* genes have been used to characterize the diversity of clover root-nodule samples, successfully identifying genospecies A to E [81]. We assessed the sequence diversity of these two amplicons across all 440 genomes. Because HTAS relies on high-throughput sequencing, the length of the amplicons is limited (in this case, 251 informative nucleotides for *recA,* 254 for *rpoB*). Nevertheless, most strains can be identified to the correct genospecies (Appendix A), although there are a few ambiguous sequences, particularly in the F-clade (gsN-R).

### 3.7. 16S rRNA Sequence Is Not Indicative of Genospecies

The sequence of the small-subunit ribosomal RNA (16S rRNA) gene has been a valuable marker for bacterial taxonomy that has helped to establish the relationships among higher taxa. However, it is often too conserved to provide useful discrimination among closely related species. In fact, the full-length 16S sequences of all five type strains within the Rlc (those of *R. leguminosarum, R. laguerreae, R. sophorae, R. ruizarguesonis and “R. indicum”*) are identical. The *R. anhuiense* sequence is also identical, and even that of the more distantly related *R. acidisoli*. It would be reasonable to expect that there would be no variation in 16S sequence within the Rlc, but this is far from the truth. This ‘standard’ sequence is indeed the most frequent, found in 286 of the 440 genomes (including all *R. anhuiense*), but there is variation in several parts of the 16S rRNA, as indicated in Figure 3 and detailed in Appendix A. Kumar et al. [29] reported a single nucleotide that varied in genospecies A to E (T in gsA and gsB, C or A in gsC, A in gsD, C in gsE), and this polymorphism is widespread across the genospecies of the Rlc. The fourth possible nucleotide (G) is also found. While there is some association with genospecies (e.g., all gsB strains have T, all gsM have G), every variant is found in more than one genospecies. In the 16S rRNA secondary structure, this polymorphic site is an unpaired base in a loop (position 1076 in the standard sequence). Another polymorphism that is shared by gsM and some, but not all, strains in the F-clade affects a pair of bases in a stem (positions 948 and 961) that are T and A in the standard sequence, C and G in the variant. A third variant is found in eleven strains of gsO, gsP and gsQ. This is the insertion of an intervening sequence (IVS) 78 nt in length in place of the normal 4-base loop at positions 73–76. This IVS has been described previously in a number of Rlc strains, including three of those reported here [95]. Two strains have a single-nucleotide variant within the IVS (white triangles in Figure 3). There are a number of other 16S sequence variants that are confined to one or two strains each. Excluding five single-nucleotide variants that were each found only in a single strain, and might represent sequencing errors, the combined effect of all this polymorphism is that we found 18 distinct 16S sequences within the Rlc. Most of the variation is, however, distributed across several genospecies and hence not useful for identifying genospecies. The sequences characterized here are those with the best match to the type strain sequence. Since *Rhizobium* genomes normally have three copies of the ribosomal RNA operon, within-strain variation is possible in principle, but cannot be investigated thoroughly with this set of genomes because most are not fully assembled and include only a single consensus 16S sequence.

### 3.8. Nodulation Specificity Is Not a Useful Taxonomic Character

Host specificity for nodulation was abandoned long ago as a taxonomic character because of the evidence for widespread mobility of nodulation genes within and between species [42,43,45,46,96]. This is very evident from the Rlc genome sequences. Nearly all the strains were originally isolated from legume root nodules, although some were not, but even among nodule isolates there are a few genomes that have no nodulation genes, probably because of loss of the symbiosis plasmid before sequencing. In some cases, e.g., SM168B [21], the original culture was shown to nodulate and had nodulation genes detectable by PCR. A colored circle in Figure 3 shows the symbiovar of each strain, as determined from the sequence of its *nodC* gene. Symbiovar *viciae* strains (nodulating plants in the genera *Vicia, Lathyrus, Pisum, Lens* and *Vavilovia*) and *trifolii* strains (nodulating *Trifolium*) are well mixed in most genospecies; even closely related strains may differ in symbiovar. The distribution is not completely random—all the gsA strains are *trifolii* and most F-clade strains are *viciae*, for example—but it is clear that symbiovar is not a useful character for distinguishing genospecies.

The nodulation genes of the three symbiovars, *viciae*, *trifolii* and *phaseoli*, are very distinct in sequence, but there is also polymorphism within each symbiovar. For example, the phylogeny of *nodC* (Appendix A) shows multiple alleles within each of the three symbiovar-specific clades. Symbiovar *trifolii* shows the pattern reported previously [21], in that most alleles are confined to a single genospecies, but a few are more promiscuous, showing high levels of introgression. Intriguingly, most symbiovar *viciae* alleles are found in more than one genospecies (Appendix A), suggesting that *viciae* symbiosis genes may be more mobile than their *trifolii* relatives. We note, however, that the *viciae* isolates have been sampled from a more diverse set of studies, locations and hosts than the *trifolii* isolates, which may be reflected in the observed patterns of diversity.

### 3.9. Genospecies Have Distinct Plasmid-Borne Sequences

Rlc genomes typically have a chromosome of approximately 5 Mb plus four to six plasmids totaling another 2.5–3 Mb. Thus, approximately one-third of the genome is extrachromosomal. The larger plasmids are chromids that resemble the chromosome in G + C composition and in having a largely stable set of genes [97], while the smaller plasmids mostly carry accessory genes, i.e., genes that are present in some strains but not others, and these genes tend to have a lower G + C content [66]. The bac120 core genes are all chromosomal, and ANI will be dominated by chromosomal matches, but we wondered whether plasmid-borne genes were also characteristic of their genospecies. We sorted the scaffolds of each genome into chromosomal and nonchromosomal (i.e., plasmid) compartments and calculated ANI separately for each compartment. Figure 7 shows that the ANI of plasmid DNA is strongly correlated with that of chromosomal DNA, but that the values are usually lower and are more variable. We can conclude that the fraction of the plasmid-borne genes that is in common between two strains has a strong genospecies-specific signature, but tends to evolve in sequence faster than the chromosome. This may reflect lower selective constraints, as most essential genes are on the chromosome.

### 3.10. Genospecies Have Distinct Gene Complements

While variation in the sequence of shared genes may contribute to functional divergence of the genospecies, it is likely that important differences are also conferred by genes that are present in some strains but absent from others. We therefore sorted all the protein-coding genes found in strains of the Rlc into sets of orthologs, and explored the distribution of these ortholog sets across the genospecies of the Rlc. The level of gene sharing is higher between strains within a genospecies than between those in different genospecies (Appendix A), confirming a previous study that was restricted to five genospecies [21]. This demonstrates that each genospecies does, indeed, have a characteristic set of accessory genes. Gene sharing is strongly correlated with ANI both within and between genospecies (Figure 8). The fact that more closely related strains share more accessory genes could be explained by vertical inheritance of these genes from a common ancestor, but it is also plausible that the rates of horizontal gene transfer are higher between more similar strains and gene sharing is a more dynamic process. Phylogenetic analysis of the shared genes would shed light on this question, but this is beyond the scope of the present study.

## 4. Conclusions

We have shown that the *R. leguminosarum* species complex (Rlc) forms a distinct clade and is clearly separated from other species in the genus by a long branch in the core gene phylogeny and a gap in ANI values (Figure 1 and Figure 2). We have also shown that the Rlc can be subdivided into 18 clusters that are sufficiently different to be considered separate genospecies, plus a further 7 single strains that might be the first representatives of additional genospecies (Figure 3, Figure 4 and Figure 5). It is not our purpose here to propose formal species names for these genospecies, but the evidence provided here is certainly a good starting point for such proposals in the future.

Five years ago, many bacterial species descriptions were still being published without the genome sequence of the proposed type strain, but a genome sequence is now almost universally required. Often, though, taxonomists are not fully exploiting the power of genomes. The first important point is that phylogenetic inferences should be based on the sequences of a large number of core genes, such as the set of 120 genes used here. In the past, taxonomists have often relied on just a couple of kilobases of sequence from two or three housekeeping genes, but our analyses show just how unreliable this can be. It is clear that there has been substantial introgression of alleles between genospecies affecting housekeeping genes such as *recA, atpD, rpoB* and *gyrB*, such that a single genospecies may have alleles that are not closely related (Figure 6 and Appendix A). In most cases, this introgression appears to have been ancient and followed by sequence divergence, since it is rare to find exactly the same allele in different genospecies. As a result, the sequence of a single gene may identify a known genospecies unambiguously, but give a very misleading indication of the phylogenetic relationships among strains.

A second point is that insightful taxonomy cannot be based entirely on type strains. If we had confined our study to type strains, we would have had just five data points and would have learned nothing except that *R. laguerreae*, *R. sophorae* and *“R. indicum”* were more closely related to each other than to *R. leguminosarum* and *R. ruizarguesonis*. Yet, when each of those species was established, it was based on a handful of strains that were compared only with the existing type strains. The consequence is most dramatically illustrated in the case of *R. ruizarguesonis*. It was described, perfectly correctly, using four similar strains [58]. However, a search of GenBank would have revealed that there were already well over a hundred available genomes of the same species (gsC), greatly expanding the known diversity and geographic range of the species. Until recently, taxonomists have tended to consider only a small number of strains when describing a new species, limited by the practicality of obtaining strains and characterizing them in the laboratory. The increasing availability of large numbers of genome sequences gives access to a much richer picture of the extent of variability within each species, potentially leading to more robust and useful species definitions.

There have been recent initiatives to harness the power of genome sequences in the service of bacterial taxonomy. Two are particularly noteworthy for their ambitious scope, covering all bacteria and archaea, and for their contrasting philosophies. The Type Strain Genome Server (TYGS) aims to identify any submitted genome by comparison to the genomes of all published type strains [30]. Strains that do not fit within the species that have been formally named are simply returned as ‘potential new species’. By contrast, the developers of the Genome Taxonomy Database (GTDB) take a more inclusive approach, using a 95% ANI threshold to place all available genomes into species-level groups [20].The result is a parallel nomenclature that sometimes diverges from the formal taxonomy but is arguably a better and more consistent reflection of biological reality. Almost two-thirds of the species-level taxa in GTDB have no formal name yet. These include, of course, some of the genospecies we have identified in the Rlc. Our analyses suggest that, within the Rlc, the natural boundaries are mostly rather higher than 95% ANI, centered at approximately 96%, so GTDB does not recognize some of the splits we have discussed. Of course, it cannot be expected that a fixed threshold will coincide with natural species boundaries across the whole of the prokaryotes. The rates of sequence evolution vary across lineages [13], as do population sizes, ecological opportunities, and so on. We do not fully understand the process of speciation in bacteria, but a stable and useful taxonomy needs to reflect the resulting pattern of clusters and gaps that is being revealed by genome sequencing. The mechanisms that create and maintain these gaps in bacterial ‘genomic space’ could be genetic, ecological, or both. These ideas have been explored from various perspectives [25,98,99,100,101,102], and we can expect that high-throughput genome sequencing will contribute to a new level of understanding of bacterial evolution and speciation. For our purpose here, though, it is enough to note that bacterial species exist in nature, and genome sequencing enables us to discover and describe them.

## Figures and Tables

**Figure 2 genes-12-00111-f002:**
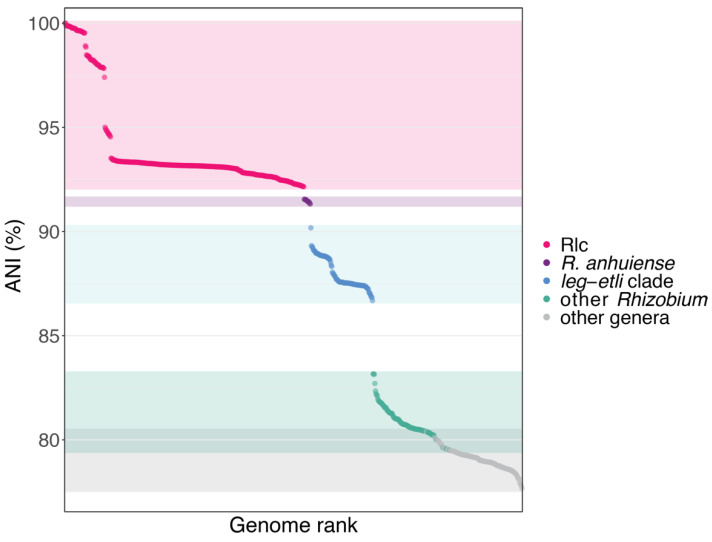
Average nucleotide identity (ANI) to the type strain of *R. leguminosarum* (USDA 2370T) of 766 of the genomes shown in Figure 1, in rank order of ANI. Some more distant genomes were omitted because ANI could not be calculated.

**Figure 3 genes-12-00111-f003:**
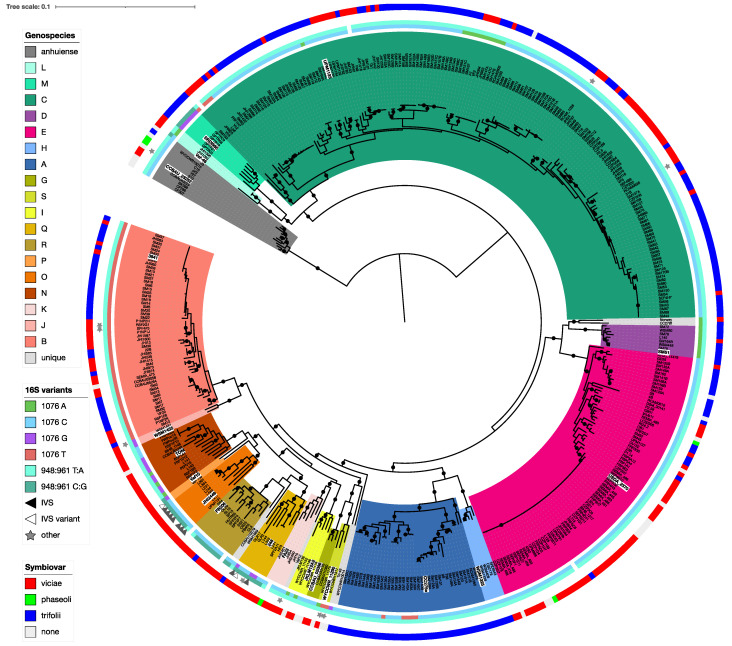
Phylogeny of the Rlc based on 120 core genes. The tree is rooted using *R. anhuiense* as the outgroup. The 18 potential genospecies are indicated by colored segments. Inner circles and symbols indicate variation in the sequence of the 16S rRNA gene (see text for details). The outer circle indicates the symbiovar inferred from the sequence of Nod genes (if any are present in the assembly). Black dots indicate branches with 100% bootstrap support. The representative strain for each genospecies is highlighted in white. An interactive version is available at https://itol.embl.de/shared/rhizobium.

**Figure 4 genes-12-00111-f004:**
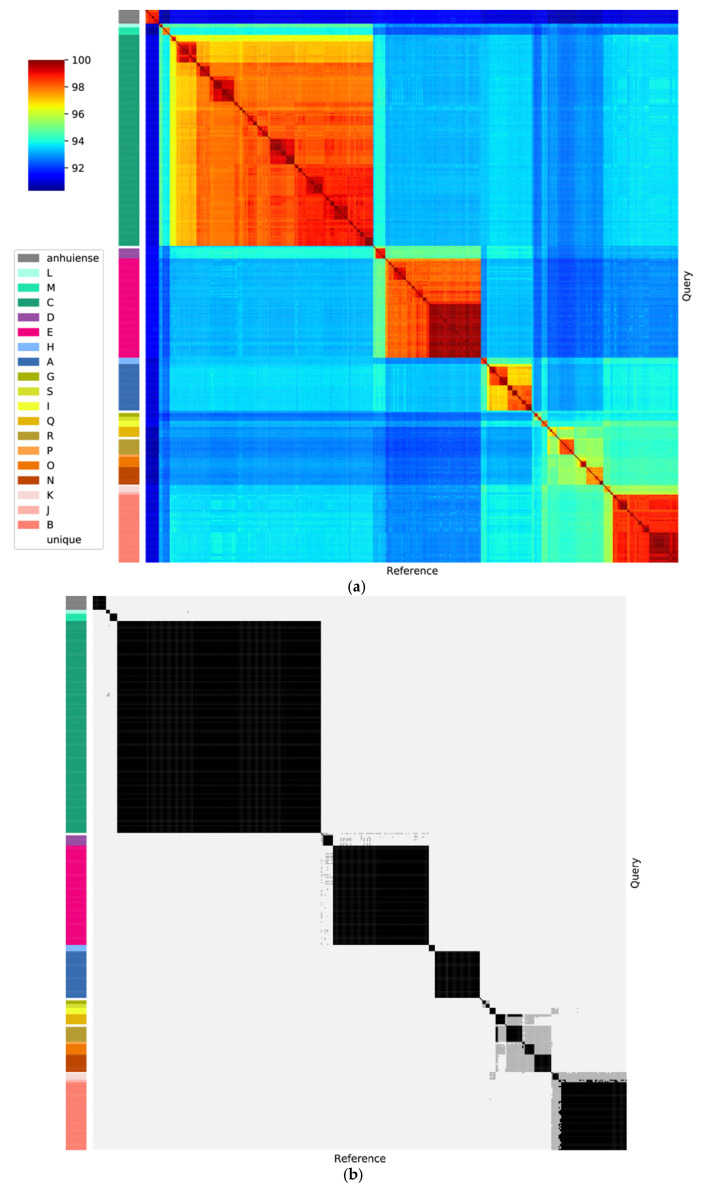
Pairwise average nucleotide identity (ANI) between all genomes in the Rlc and *R. anhuiense*. The bars at the left indicates genospecies of each genome. (**a**) Continuous color scale. (**b**) The same data, but with thresholds to indicate values over 95% (grey) or over 96% (black).

**Figure 5 genes-12-00111-f005:**
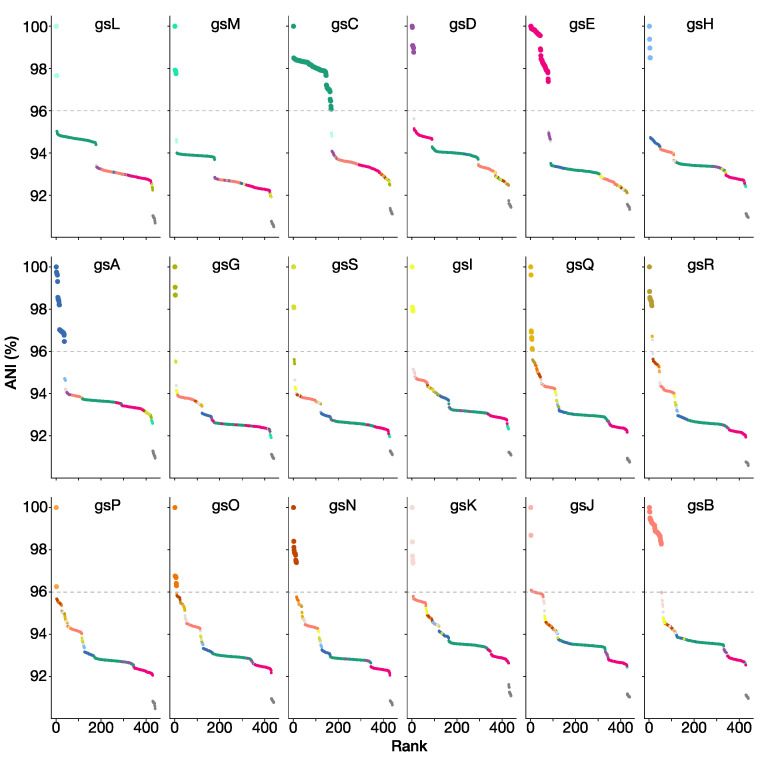
ANI plots using the representative strains of each of the 18 genospecies as reference. Points are colored by genospecies. Strains that are conspecific with the reference are shown with larger symbols. Dashed line at 96% ANI indicates conventional species boundary.

**Figure 6 genes-12-00111-f006:**
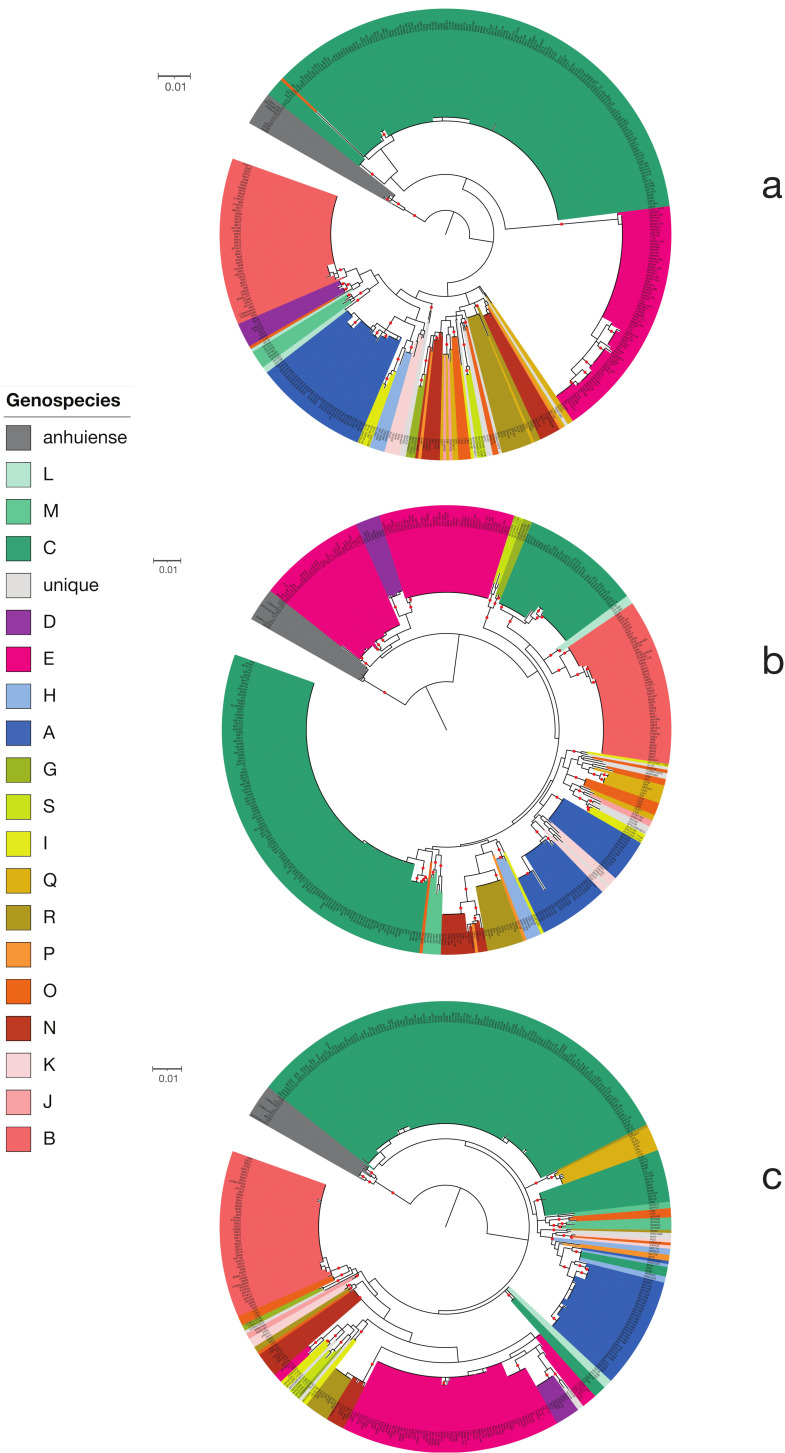
Phylogenies of individual housekeeping gene amplicons. (**a**) *atpD*. (**b**) *gyrB*. (**c**) *recA*. Colors indicate genospecies. Red dots indicate bootstrap support >70%. Scale bars indicate 1% sequence divergence. Interactive versions are available at https://itol.embl.de/shared/rhizobium.

**Figure 7 genes-12-00111-f007:**
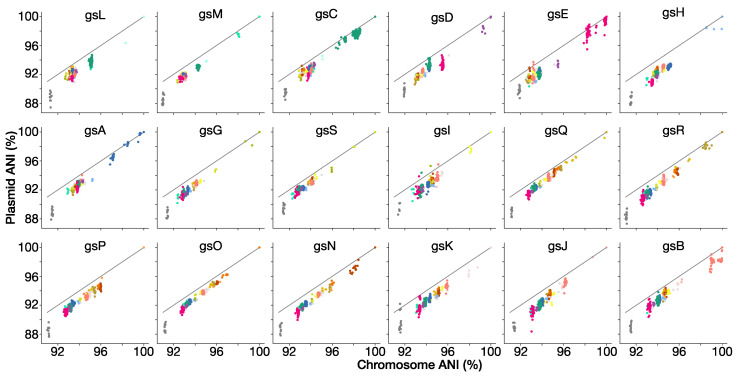
ANI of plasmid and chromosome genome compartments. The plasmid compartment includes chromids. Each point represents the values for the two compartments in a single pairwise comparison between genomes. Each panel shows values for every strain (colored by genospecies) compared to the representative strain for the indicated genospecies.

**Figure 8 genes-12-00111-f008:**
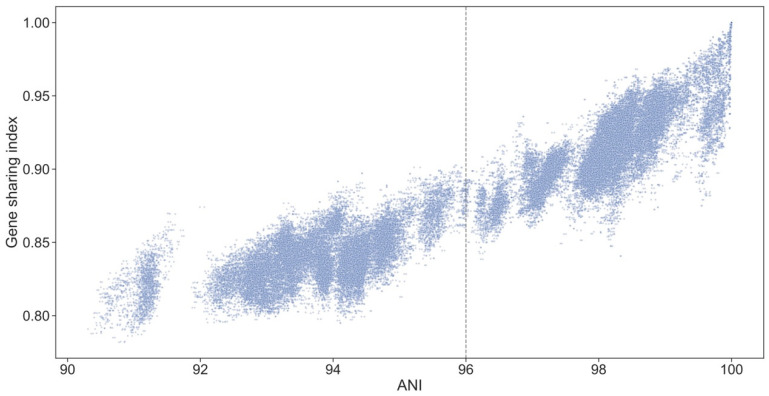
Sharing of accessory genes between strains compared to ANI. The gene sharing index is based on the number of orthogroups shared between two strains, normalized by total orthogroup content, for selected comparisons. See Appendix A. Dashed line at 96% ANI indicates approximate genospecies boundary.

**Table 1 genes-12-00111-t001:** Equivalence between our genospecies and the species-level taxa defined in the Genome Taxonomy Database (https://gtdb.ecogenomic.org).

Genospecies or Strain	GTDB Species
*R. anhuiense*	s__Rhizobium anhuiense
L	s__Rhizobium leguminosarum_D
M	s__Rhizobium leguminosarum_I
C	s__Rhizobium leguminosarum_C
D + CC278f + Norway	s__Rhizobium leguminosarum_K
E	s__Rhizobium leguminosarum
H	s__Rhizobium leguminosarum_J
A	s__Rhizobium leguminosarum_E
WYCCWR10014	s__Rhizobium sp001657485
Tri-43	s__Rhizobium leguminosarum_M
G	not represented
S	not represented
I	not represented
Q, WSM1689, CCBAU10279, R, P, O, N	s__Rhizobium laguerreae
Vaf12	s__Rhizobium sp005860925
K, J, B	s__Rhizobium leguminosarum_L

## Data Availability

All genome sequences used in this study are openly available in the INSDC databases (http://www.insdc.org/). Data analysis scripts are available at https://github.com/jpwyoung/Rlc and phylogenetic trees at https://itol.embl.de/shared/rhizobium.

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
