# Peer review of "Defining the Rhizobium leguminosarum Species Complex"

_genes, 2021, doi:10.3390/genes12010111_

Round 1
Reviewer 1 Report
This manuscript has investigated the genetic and phylogenetic diversity of the 429 publicly available genome sequences that are currently designated as belonging to the species Rhizobium leguminosarum (designated as Rlc). Earlier studies on a smaller number of strains for R. leguminosarum have shown that these strains form several distinct clades (referred to as genospecies)in phylogenetic trees. However, this work has comprehensively investigated the relationships among different strains by means of constructing robust phylogenetic trees based on concatenated sequences of 120 core genes that are commonly shared by different strains and by also determining pairwise average nucleotide identity (ANI) between all genomes. The results from these analyses provide good evidence for the existence of 18 distinct clades (designated as genospecies) within the Rlc, which are separated by a distinct gap in ANI values, usually at around 96% ANI. Several of these clades also correspond to the grouping of these strains in the Genome taxonomy database. The results presented suggest that the observed clades upon further characterization could be candiates for distinct species. The authors note that the observed diversity among the Rlc strains could not be deciphered by means of 16S rRNA trees or trees based on limited genetic information. Thus, analysis based on genome sequences is very useful for reliable determination of bacteria diversity and for taxonomic inferences.
Comments; It is a comprehensive study examining the phylogenetic diversity of Rhizobium leguminosarum strains. The inferences from this study, which are well supported by the presented data, are important for understanding the extensive diversity that exists within this group of organism. Additionally, this work also illustrates and emphasize that analysis based on genome sequences is very useful and necessary for reliable determination of bacteria diversity and taxonomic inferences. The manuscript is clearly written and the data presentation is clear.
Author Response
We thank the reviewer for these supportive comments, which do not raise any issues that require attention.
Reviewer 2 Report
The manuscript "Defining the Rhizobium leguminosarum species complex" by Young et al. describes a taxonomic study of the Rhizobium leguminosarum species complex that was performed using genome-level analyses. The R. leguminosarum species complex is incredibly diverse, and previously, its taxonomy had been poorly defined. The presented work addresses this gap in the literature. I believe this work is an important and much needed study that will be of interest to the rhizobium community, and the genomic and taxonomic fields more broadly. I enjoyed reading the manuscript and through that it was well written on the whole. Nevertheless, I do have several suggestions for the Authors to consider.
My main comment relates to the choice of software for phylogenetic tree reconstruction. Most phylogenies in the manuscript are prepared with FastTree. While notable for its speed, benchmarking studies (e.g., doi:10.1093/molbev/msx302) have shown that FastTree underperforms in accuracy compared to other software. I suggest the Authors reperform the phylogenetic analyses with a program such as IQ-tree or RAxML. If the Authors do not have ready access to local computational resources for these analyses, they can be performed for free on the CIPRES Web Portal (http://www.phylo.org), with the speed of the analysis quite rapid for datasets of similar sizes to those used throughout this study.
Lines 156-158: I believe there is a word (or words) missing from this sentence, potentially before the phrase “…decided which was true…”
Line 224: If there is a particular reason for choosing NodA, NodC, and NodD (and not, for example, NodB), I suggest the Authors note the reason in the methods.
Line 227: Some rhizobia contain multiple copies of some nod genes, including multiple copies of nodD. Is this true for any strains within the Rhizobium leguminosarum species complex? And if so, how were redundancies treated in the analysis?
Lines 245-252: While I trust that this analysis effectively (and correctly) separated the contigs into chromosomal and non-chromosomal segments, do the Authors have any data validating the approach? For example, mapping the contigs of a draft genome to a closely related finished genome, and checking what percentage of reads mapping to the chromosome or extra-chromosomal replicons are correctly called by the pipeline.
Lines 299-300: This statement seems to imply that "R. album" NS-104 is not part of the genus Rhizobium; however, the corresponding branch in Figure 1 appears to be green, suggesting it is part of the genus Rhizobium. Could the Authors clarify if this strain is part of the genus Rhizobium?
Figures 3 & 6: I realize it may be difficult given the size of the phylogenies, but it would be ideal to represent bootstrap values on the tree.
Lines 413-424: I suggest the Authors consider giving an example of a technical reason.
Line 474: I believe "more distant that is usual" should be "more distant than is usual".
Lines 495-497: Perhaps I am misunderstanding, but these statements appear contradictory; how can all 18 genospecies be resolved if two of the genospecies did not form monophyletic groups?
Lines 518-520: I suggest the Authors consider expanding on this point (if it remains true in phylogenies prepared with RAxML or IQ-tree) as I think it is an important point, and I think it could be made more clear that this statement is referring to the relationships between the genospecies differ between phylogenies.
Lines 603-604: I realize the Authors may be unable to answer this question; however, do the Authors have any insight into whether the correlation between the two components is driven primarily by the chromids?
Line 602: In some locations the extrachromosomal compartment is referred to as extrachromosomal, and in others, it is referred to as plasmid. I suggest the Authors consider using "extrachromosomal" throughout, as this compartment includes both plasmid and chromid genes.
Lines 617-618: I'm not sure that the data, as presented, support the statement that each genospecies has a characteristic set of accessory genes. Just because strains within a genospecies share more genes, doesn't necessarily mean that those genes are not also in other genospecies. Have the Authors examined what percent of accessory genes are found specifically within a single cenospecies?
Author Response
The manuscript "Defining the Rhizobium leguminosarum species complex" by Young et al. describes a taxonomic study of the Rhizobium leguminosarum species complex that was performed using genome-level analyses. The R. leguminosarum species complex is incredibly diverse, and previously, its taxonomy had been poorly defined. The presented work addresses this gap in the literature. I believe this work is an important and much needed study that will be of interest to the rhizobium community, and the genomic and taxonomic fields more broadly. I enjoyed reading the manuscript and through that it was well written on the whole. Nevertheless, I do have several suggestions for the Authors to consider.
>>>Thank you for this expert and thoughtful review. We have responded to all the points raised, and the manuscript has been substantially improved as a result.
My main comment relates to the choice of software for phylogenetic tree reconstruction. Most phylogenies in the manuscript are prepared with FastTree. While notable for its speed, benchmarking studies (e.g., doi:10.1093/molbev/msx302) have shown that FastTree underperforms in accuracy compared to other software. I suggest the Authors reperform the phylogenetic analyses with a program such as IQ-tree or RAxML. If the Authors do not have ready access to local computational resources for these analyses, they can be performed for free on the CIPRES Web Portal (http://www.phylo.org), with the speed of the analysis quite rapid for datasets of similar sizes to those used throughout this study.
>>>Thank you for this reference (which we have cited in the Methods) and the heads-up about CIPRES, which we have used to create new trees with RAxML-NG. We have replaced Figure 3 with a new phylogeny using RAxML-NG and moved the original fasttree version to Supplementary Figures (new Figure S1). The conclusions are unaltered: all 17 genospecies that had strong support by fasttree have 100% bootstrap support by RAxML-NG. Relationships among genospecies are largely unaltered, except that gsK has moved slightly – but neither position was strongly supported in either tree. The two strains we designated gsP do not form a clade in the RAxML-NG tree – but we had already concluded that there was no support for this clade in the fasttree tree either. We have also replaced the other trees with RAxML-NG versions, except Figure 1, and updated the Methods accordingly.
Lines 156-158: I believe there is a word (or words) missing from this sentence, potentially before the phrase “…decided which was true…”
>>> The sentence seems complete to us: "They also discovered …, decided …, and defined …". For greater clarity, we have added "of these" in "decided which of these was …".
Line 224: If there is a particular reason for choosing NodA, NodC, and NodD (and not, for example, NodB), I suggest the Authors note the reason in the methods.
>>>We first looked at nodD and nodA, representing two operons, but there was not complete agreement, so we added nodC (which showed us that nodA was the outlier, being present in some SEMIA strains that did not have nodD or nodC). To save explanation, we have now added nodB, which agrees completely with nodD and nodC. Incidentally, we checked nodA in a SEMIA strain assembly – it is surrounded by hypothetical and pseudogenes in a short contig that also carries nitrogenase genes, but no other nod genes.
Line 227: Some rhizobia contain multiple copies of some nod genes, including multiple copies of nodD. Is this true for any strains within the Rhizobium leguminosarum species complex? And if so, how were redundancies treated in the analysis?
>>>This is true, but when multiple copies exist, they are still symbiovar-specific, so this does not affect the symbiovar classification. For the nodC phylogeny, we used just the copy with the best hit to the reference sequence.
Lines 245-252: While I trust that this analysis effectively (and correctly) separated the contigs into chromosomal and non-chromosomal segments, do the Authors have any data validating the approach? For example, mapping the contigs of a draft genome to a closely related finished genome, and checking what percentage of reads mapping to the chromosome or extra-chromosomal replicons are correctly called by the pipeline.
>>>It is clear that much of the ANI signal comes from core genes on the chromosome, but we wanted to see whether scaffolds that were not chromosomal also showed a similar signal. To identify scaffolds (not reads) as putatively chromosomal, we used a set of 3215 genes that we have proviously shown to be chromosomal, and always in the same order, in a small set of PacBio genomes (Cavassim et al. 2020). Of these genes, 2565 were found in all 440 genomes. All these genes had hits in the chromosome of all 22 fully-assembled Rlc genomes, though three Russian strains also had a duplication of four adjacent genes on a plasmid. We counted a scaffold as chromosomal if it had at least one best hit, which would have led to this plasmid being misclassified in the chromosomal fraction. On the other hand, if we required more hits per scaffold, some smaller chromosomal scaffolds would be misclassified. We reran the classifier requiring at leat 5 hits per scaffold – this led to changes in 202 of the 440 genomes, and a decrease in average "chromosome size" of about 1% (50kb). This is not a huge change, so we think our approach is reasonably robust. We have added a brief summary of this.
Lines 299-300: This statement seems to imply that "R. album" NS-104 is not part of the genus Rhizobium; however, the corresponding branch in Figure 1 appears to be green, suggesting it is part of the genus Rhizobium. Could the Authors clarify if this strain is part of the genus Rhizobium?
>>> The proposed name is in the genus Rhizobium, and there is no other genus that is any closer. We have added a sentence: " Although described as a species within Rhizobium, its distance from all other species in the genus (Figure 1) suggests that this needs reconsideration."
Figures 3 & 6: I realize it may be difficult given the size of the phylogenies, but it would be ideal to represent bootstrap values on the tree.
>>>We have included symbols indicating either 100% or >70% bootstrap support in the new phylogenies obtained with RAxML-NG. Full bootstrap values for all trees are provided in the online iTOL versions of the phylogenies, available at the URL indicated in the legends. We have added a mention of bootstraps to the legends.
Lines 413-424: I suggest the Authors consider giving an example of a technical reason.
>>>The most likely technical reason would be a problem with the quality of the assembly, but we have not examined these genomes critically so we would not like to cast doubt on them without evidence.
Line 474: I believe "more distant that is usual" should be "more distant than is usual".
>> Typo corrected.
Lines 495-497: Perhaps I am misunderstanding, but these statements appear contradictory; how can all 18 genospecies be resolved if two of the genospecies did not form monophyletic groups?
>>>A given sequence can be assigned unambiguously to a single species if it is only found in one species, even if the set of species-specific sequences is not monophyletic. We have changed "resolve" to "classify strains of", which we hope is clearer.
Lines 518-520: I suggest the Authors consider expanding on this point (if it remains true in phylogenies prepared with RAxML or IQ-tree) as I think it is an important point, and I think it could be made more clear that this statement is referring to the relationships between the genospecies differ between phylogenies.
>>>We have replaced Figure S2 with a RAxML-NG version, which still shows some genospecies in very different places compared to the 120-gene tree. We agree that this is the important point, and have added a couple of extra sentences as follows. "While most (though not all) of the genospecies form single clades, the relationships among genospecies are sometimes radically different. For example, the related genospecies gsC, gsL and gsM are very distant in the three-gene tree, as are the close pair gsD and gsE."
Lines 603-604: I realize the Authors may be unable to answer this question; however, do the Authors have any insight into whether the correlation between the two components is driven primarily by the chromids?
>>>It probably is, because the majority of the shared nonchromosomal genes will be on the chromids, but as the vast majority of the genomes are not fully assembled we cannot test this robustly.
Line 602: In some locations the extrachromosomal compartment is referred to as extrachromosomal, and in others, it is referred to as plasmid. I suggest the Authors consider using "extrachromosomal" throughout, as this compartment includes both plasmid and chromid genes.
>>>A chromid is a special kind of plasmid (as explained in section 3.9) , but we understand the reviewer's point and have added text to make it clear that we include chromids when we mention plasmids (legend to Figure 7).
Lines 617-618: I'm not sure that the data, as presented, support the statement that each genospecies has a characteristic set of accessory genes. Just because strains within a genospecies share more genes, doesn't necessarily mean that those genes are not also in other genospecies. Have the Authors examined what percent of accessory genes are found specifically within a single cenospecies?
>>>We would argue that a particular combination of genes can be characteristic of a genospecies even if the genes are not unique to the genospecies. Figure S5 indicates a clear drop in the fraction of shared genes at the genospecies boundaries. It would be good to identify genospecies-specific genes, but this requires a comprehensive analysis of orthology, and so far we have not found ortholog-identification software that scales well to the large number of genomes in our study. Cavassim et al. (2020) did this for a smaller number of genomes in genospecies A to E; here, we can only present a preliminary analysis that indicates that their observations can probably be extended to this larger and more diverse sample. We have moved up the sentence about a "characteristic set" to come directly after the mention of Figure S5, which provides the visual evidence for this.
Round 2
Reviewer 2 Report
The Authors have appropriately responded to all of my comments, and I look forward to seeing the final formatted version of the interesting manuscript.